# Migrating birds optimization-based feature selection for text classification

Cem Kaya[1,2], Zeynep Hilal Kilimci[3], Mitat Uysal[1] and Murat Kaya[4]

[1] Department of Software Engineering, Dogus University, Istanbul, Turkey
[2] Scientific and Technical Research Council of Turkiye (TUBITAK), Kocaeli, Turkey
[3] Department of Information Systems Engineering, Kocaeli University, Kocaeli, Turkey
[4] Department of Computer Programming, Acibadem University, Istanbul, Turkey

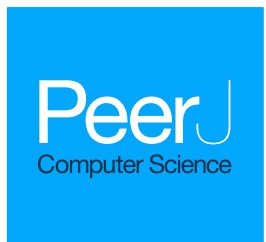

Corresponding author
Cem Kaya, cem.kaya@tubitak.gov.tr

## ABSTRACT

Text classification tasks, particularly those involving a large number of features, pose significant challenges in effective feature selection. This research introduces a novel methodology, MBO-NB, which integrates Migrating Birds Optimization (MBO) approach with naïve Bayes as an internal classifier to address these challenges. The motivation behind this study stems from the recognized limitations of existing techniques in efficiently handling extensive feature sets. Traditional approaches often fail to adequately streamline the feature selection process, resulting in suboptimal classification accuracy and increased computational overhead. In response to this need, our primary objective is to propose a scalable and effective solution that enhances both computational efficiency and classification accuracy in text classification systems. To achieve this objective, we preprocess raw data using the Information Gain algorithm, strategically reducing the feature count from an average of 62,221 to 2,089. Through extensive experiments, we demonstrate the superior effectiveness of MBO-NB in feature reduction compared to other existing techniques, resulting in significantly improved classification accuracy. Furthermore, the successful integration of naïve Bayes within MBO offers a comprehensive and well-rounded solution to the feature selection problem. In individual comparisons with Particle Swarm Optimization (PSO), MBO-NB consistently outperforms by an average of 6.9% across four setups. This research provides valuable insights into enhancing feature selection methods, thereby contributing to the advancement of text classification techniques. By offering a scalable and effective solution, MBO-NB addresses the pressing need for improved feature selection methods in text classification, thereby facilitating the development of more robust and efficient classification systems.

## INTRODUCTION

Text classification involves the automatic assignment of predefined categories or labels to text documents, making it a supervised machine learning task (*Sebastiani, 2002*; *Aggarwal, 2015*; *Hotho, Nürnberger & Paaß, 2005*; *Su, Zhang & Xin, 2006*; *Joachims, 1998*). The process encompasses several stages, including data preprocessing (*Uysal & Gunal, 2014*), feature selection (*Liang et al., 2017*), model training, and evaluation (*Yu, 2008*). During

data preprocessing, the input text undergoes cleaning and transformation into a numerical representation, such as bag-of-words, Term Frequency-Inverse Document Frequency (TF-IDF), or word embeddings, enabling the application of machine learning algorithms. Feature selection aims to extract the most pertinent information from the text while discarding irrelevant data. Model training entails the selection of an appropriate algorithm, parameter optimization, and evaluation of its generalization performance using a held-out set of labeled examples. Commonly utilized machine learning algorithms for text classification include naïve Bayes, Decision Trees, support vector machines (SVM), and neural networks (*Pranckevicius & Marcinkevicius, 2017*). Text classification represents a crucial problem in natural language processing with widespread applications in diverse domains (*Kowsari et al., 2019*), including sentiment analysis (*Pang & Lee, 2008*), spam detection (*Cormack, 2008*), topic modeling (*Spasic & Nenadic, 2020*; *Weng et al., 2010*), and language identification (*Sebastiani, 2002*).

Feature selection constitutes a crucial step in text classification, aiming to ascertain the most relevant features suitable for categorizing text documents into distinct classes (*Yang & Pedersen, 1997*; *Sebastiani, 2002*). Generally, it entails isolating a subset of features from a larger pool, deemed most pertinent for classification purposes (*Chandrashekar & Sahin, 2014*; *Kowsari et al., 2019*; *Chen et al., 2009*). The objective of feature selection is to enhance the classification model's accuracy while mitigating computational complexity and the risk of overfitting. Numerous techniques are available for feature selection in text classification, encompassing filter methods (*Sánchez-Maroño, Alonso-Betanzos & Tombilla-Sanromán, 2007*; *Labani, Moradi & Jalili, 2020*), wrapper methods (*Hu et al., 2015*; *Ghareb, Bakar & Hamdan, 2016*), and embedded methods (*Moslehi & Haeri, 2020*; *Stein, Jaques & Valiati, 2019*).

Heuristic optimization emerges as an alternative approach for tackling optimization dilemmas, rooted in intelligent search and problem-solving strategies, rather than strict mathematical models or exact algorithms (*Kennedy & Eberhart, 1995*; *Rardin & Uzsoy, 2001*; *Mavrovouniotis, Li & Yang, 2017*; *Dorigo, Birattari & Stutzle, 2006*). Heuristic optimization algorithms are devised to swiftly explore a problem's search space and procure satisfactory solutions, albeit without guaranteeing optimality (*Talbi, 2009*; *Blum & Roli, 2003*; *Reeves, 1993*; *Kendall, 2014*). Typically, these approaches involve generating a pool of potential solutions, assessing their quality *via* fitness or objective functions, and iteratively refining them using various search or optimization methodologies (*Michalewicz & Fogel, 2013*; *Lee & El-Sharkawi, 2008*; *Glover, 1977*). Diverse techniques are employed to navigate the search space, including local search, simulated annealing (*van Laarhoven & Aarts, 1987*), genetic algorithms, Particle Swarm Optimization (PSO) (*Kennedy & Eberhart, 1995*; *Abualigah, Khader & Hanandeh, 2018*), ant colony optimization (*Dorigo, Birattari & Stutzle, 2006*), and numerous others (*Grefenstette, 1993*; *Geem, Kim & Loganathan, 2001*; *Yang & Deb, 2009*; *Hansen & Mladenović, 1999*). Heuristic optimization methods find application in complex problems where exact solutions are elusive or where the search space is extensive. Examples of such problems encompass vehicle routing (*Chang & Chen, 2007*), resource allocation, scheduling (*Guo et al., 2012*),

network design (*Crainic et al., 2011*), and numerous others (*Kaveh & Talatahari, 2010*; *Mirjalili, 2016*; *Rardin & Uzsoy, 2001*; *Colorni et al., 1996*; *Juan et al., 2015*).

The realm of feature selection, particularly in scenarios with a substantial number of features, has posed challenges for conventional algorithms, underscoring the potential effectiveness of heuristic approaches. The rationale behind this choice stems from the acknowledgment of heuristics' capability, especially in circumstances where the quest for global maxima proves arduous, and local maxima suffice to tackle the complexities at hand. In this regard, Migration Birds Optimization (MBO) (*Duman, Uysal & Alkaya, 2012*) has been regarded as an emerging standout in the field, offering adept solutions to intricate problems (*Zhang et al., 2017a*, *2017b*; *Tongur & Ülker, 2016*; *Tongur, Ertunc & Uyan, 2020*). MBO, a nature-inspired optimization algorithm, draws inspiration from the collective behavior observed in bird flocks during migration, where individuals exchange information to enhance the overall group performance. In MBO, solutions are represented as individuals within a population, and the optimization process involves the iterative movement and interaction of these individuals to discover optimal or near-optimal solutions to a given problem.

In this study, a novel heuristic optimization-based feature selection technique for text classification utilizing MBO (*Duman, Uysal & Alkaya, 2012*) is proposed. To the best of the authors' knowledge, this represents the first instance of employing MBO as a feature selection algorithm in text classification. The motivation behind our study arises from the limitations observed in existing techniques for handling extensive feature sets. Traditional approaches often struggle to efficiently handle high-dimensional data, leading to suboptimal classification performance and increased computational overhead. Recognizing the need for scalable and effective solutions, we sought to develop a novel methodology that streamlines the feature selection process while improving classification accuracy. The experiment results show that the inclusion of MBO technique as a feature selection model performs remarkable classification results when compared to state-of-the-art studies.

The contribution of this article can be listed as follows:

- Novel methodology (MBO-NB): Introducing a novel methodology named MBO-NB that integrates Migrating Birds Optimization (MBO) approach with naïve Bayes as an internal classifier to address challenges in feature selection for text classification tasks.
- Addressing challenges with large feature sets: Recognizing and addressing the challenges posed by text classification tasks with a large number of features, particularly in the context of effective feature selection.
- Improved computational efficiency: Proposing a scalable solution that enhances computational efficiency by streamlining the feature selection process, thereby reducing computational overhead.
- Enhanced classification accuracy: Demonstrating superior effectiveness in feature reduction compared to existing techniques, leading to significantly improved classification accuracy in text classification systems.

- Comprehensive solution: Offering a comprehensive solution to the feature selection problem by successfully integrating naïve Bayes within the MBO framework.
- Superior performance: Showing consistent outperformance of MBO-NB over existing techniques, such as Particle Swarm Optimization (PSO), across multiple setups, with an average improvement in classification accuracy of 6.9%.
- Insights into feature selection methods: Providing valuable insights into enhancing feature selection methods in text classification, contributing to the advancement of text classification techniques.
- Facilitating development of robust systems: Addressing the pressing need for improved feature selection methods in text classification, thereby facilitating the development of more robust and efficient classification systems.

The rest of the article is organized as follows: "Related Work" presents a review of related work on different feature selection techniques. "Proposed Framework" details the proposed framework and methods used in this research. "Experiments" gives the experimental setup and shares the results. "Discussions" discusses the experimental results. Finally, "Conclusions" concludes the article.

## RELATED WORK

MBO emerges as a heuristic optimization algorithm inspired by the behavior of migrating bird flocks, reflecting their collaboration and communication dynamics observed during migration. Within MBO, solutions manifest as individuals within a population, and the optimization process entails the iterative movement and interaction of these individuals to ascertain optimal or near-optimal solutions to a given problem. Owing to its recent surge in popularity, MBO has found application across various real-world optimization problems, including the land distribution problem (*Tongur, Ertunc & Uyan, 2020*), discrete problems (*Tongur & Ülker, 2016*), workers assignment balancing problem (*Zhang et al., 2019b*), flowshop scheduling problem (*Zhang et al., 2017a*; *Han et al., 2018*; *Zhang et al., 2019a*), system identification problem (*Makas & Yumusak, 2016*), credit card fraud detection problem (*Duman & Elikucuk, 2013*), and steel making-continuous casting problem (*Zhang et al., 2017b*).

In this section, a brief survey of state-of-the-art studies focused on the feature selection techniques in text classification is introduced. Literature studies on feature selection techniques in text classification are introduced by categorizing into four different approaches, namely, filter methods, wrapper models, embedded methods, and heuristic optimization-based techniques.

### Feature selection with filter methods

In *Labani, Moradi & Jalili (2020)*, a novel multi-objective algorithm called Multi-Objective Relative Discriminative Criterion (MORDC) for text feature selection is introduced. A balance between minimizing redundant features and maximizing relevance to the target class is achieved by MORDC. The solution space is explored utilizing a multi-objective evolutionary framework. The relevance of text features to the target class is evaluated by

the first objective function, while the correlation among features is assessed by the second function. Importantly, the selected features are evaluated without utilizing learning-based methods, rendering it a multivariate filter approach. Experiments are conducted on three real-world datasets, including WebKB, 20-Newsgroups, Reuters-21578, to evaluate the effectiveness of MORDC. Comparative analyses against state-of-the-art feature selection methods demonstrate that, in most cases, superior classification performance is achieved by MORDC.

In *Paniri, Dowlatshahi & Nezamabadi-Pour (2020)*, a novel multi-label feature selection method named MLACO, based on Ant Colony Optimization (ACO), is introduced by the researchers. MLACO aims to identify the most promising features in the feature space by addressing both relevance and redundancy aspects. This is achieved by incorporating unsupervised and supervised heuristic functions and conducting multiple iterations. To expedite the convergence of the algorithm, the initial pheromone of each ant is set using the normalized cosine similarity between features and class labels. It is important to note that the proposed method does not rely on any learning algorithm and can be categorized as a filter-based approach. To assess its performance, MLACO is compared against five well-known and state-of-the-art feature selection methods using the ML-KNN classifier. Experimental results conducted on commonly employed datasets including 20-Newsgroups, Bibtex, Chemistry, CS, and Cooking demonstrate the superiority of MLACO in terms of various multi-label evaluation measures and run time.

In the study (*Uysal & Gunal, 2012*), a novel filter-based probabilistic feature selection method named the distinguishing feature selector (DFS) for text classification is introduced. The proposed method is compared to well-known filter approaches such as chi-square, information gain, Gini index, and deviation from the Poisson distribution across various datasets, classification algorithms, and evaluation metrics. The experimental results clearly demonstrate that DFS achieves competitive performance when compared to the aforementioned approaches, as evidenced by its classification accuracy, dimension reduction rate, and processing time.

Network clustering is considered a foundational unsupervised technique in the realm of knowledge discovery, aimed at grouping similar nodes together without supervision or foreknowledge regarding cluster characteristics. Amidst diverse clustering methodologies, semi-supervised clustering detection emerges as a particularly promising avenue for community detection, owing to its capacity to leverage supplementary information to glean deeper insights into network topology. Nevertheless, prior endeavors face twin challenges: reliance on linear techniques for dimensionality reduction and arbitrary selection of auxiliary information, resulting in diminished efficacy of semi-supervised community detection methodologies. To address these shortcomings, an end-to-end framework termed deep semi-supervised community detection is proposed for intricate networks. A novel learning objective (*Berahmand, Li & Xu, 2023*) is introduced that integrates a semi-autoencoder augmented with a predefined pair-wise constraint matrix grounded on point-wise mutual information within the representation layer, thereby facilitating precise feature acquisition. Furthermore, in the clustering layer, a pair-wise constraint is incorporated as a term to minimize intra-cluster distances while augmenting

inter-cluster disparities. Empirical evaluations highlight the remarkable performance of the proposed approach compared to prevailing state-of-the-art community detection methodologies in complex network settings.

## Feature selection with wrapper models

In *Forman (2004)*, it is revealed that Information Gain, alongside other existing methods, fails to produce satisfactory results when applied to an industrial text classification problem. Upon further analysis, a common flaw is discovered in a large class of feature scoring methods: they tend to prioritize highly predictive features for certain classes while neglecting features crucial for discriminating challenging classes. Even in a relatively uniform text classification task, this flaw negatively impacts performance. To address this issue, solutions inspired by round-robin scheduling are proposed that effectively circumvent this flaw without resorting to computationally expensive wrapper methods. Through empirical evaluation on a set of 19 multi-class text classification tasks, the proposed method demonstrates significant improvements in performance.

In *Hu et al. (2015)*, a hybrid filter-wrapper approach for feature selection in short-term load forecasting (STLF) is introduced. The current proposition leverages the combined strengths of filter and wrapper methods. Initially, the partial mutual information (PMI) filter method is employed to eliminate irrelevant and redundant features. Subsequently, a wrapper method, employing a firefly algorithm, is applied to further reduce redundancies while maintaining the accuracy of the forecasting process. The established support vector regression model is chosen to implement this innovative hybrid feature selection scheme. To assess the efficacy of the proposed approach, real-world electricity load datasets from a North-American electric utility and the Global Energy Forecasting Competition 2012 are employed. Experimental results conclusively demonstrate the superiority of the proposed approach over comparable methodologies.

In *Günal (2012)*, a comprehensive analysis is conducted to investigate the redundancy or relevancy of text features selected by different methods in various scenarios, including different feature set sizes, dataset characteristics, classifiers, and success measures. To achieve this, a hybrid feature selection strategy is proposed, encompassing both filter and wrapper feature selection steps. The findings from the experimental investigation highlight the superior effectiveness of combining features selected from multiple methods, as opposed to relying solely on features chosen by a single selection method. Nevertheless, the composition of this feature combination is influenced by factors such as dataset characteristics, the selection of the classification algorithm, and the success measure employed.

EGA is proposed in the study (*Ghareb, Bakar & Hamdan, 2016*), through the enhancement of crossover and mutation operators. The crossover operation is executed by partitioning the chromosome (feature subset) based on the term and document frequencies of chromosome entries (features). Meanwhile, the mutation operation is performed by considering the classifier performance of the original parents and feature importance. Consequently, the crossover and mutation operations utilize valuable information rather than relying on probability and random selection. Moreover, a hybrid

approach is developed by integrating six renowned filter feature selection methods with EGA. Within this hybrid approach, EGA is applied to various feature subsets of different sizes, which are prioritized in descending order based on their importance, followed by dimension reduction. The EGA operations are specifically targeted towards the most significant features with higher ranks. The effectiveness of the proposed approach is evaluated by employing naïve Bayes and associative classification techniques on three distinct collections of Arabic text datasets. The experimental results unequivocally demonstrate the superiority of EGA over genetic algorithm (GA). Comparative analyses between GA and EGA reveal that the latter achieves superior outcomes in terms of dimensionality reduction, time efficiency, and categorization performance. Additionally, six hybrid feature selection approaches are introduced, comprising a combination of a filter method and EGA, which are applied to diverse feature subsets. The results exhibit that these hybrid approaches surpass individual filter methods in terms of dimensionality reduction, as they achieve a higher reduction rate without sacrificing categorization precision in most scenarios.

An enhanced binary grey wolf optimizer (GWO) is proposed within a wrapper feature selection (FS) approach to address Arabic text classification problems in *Chantar et al. (2020)*. The proposed binary GWO algorithm is employed as a wrapper-based FS technique. The performance of the proposed method is evaluated using various learning models, such as decision trees, K-nearest neighbour, naïve Bayes, and SVM classifiers. Three Arabic public datasets, namely Alwatan, Akhbar-Alkhaleej, and Al-jazeera-News, are utilized to assess the effectiveness of different wrapper methods based on the binary GWO algorithm. The results and analysis indicate that the SVM-based feature selection technique with the proposed binary GWO optimizer, utilizing an elite-based crossover scheme, demonstrates superior efficacy in addressing Arabic text classification problems compared to other approaches in the field.

## Feature selection with embedded methods

In *Keyvanpour, Zandian & Abdolhosseini (2022)*, an embedded approach for feature selection is adopted, utilizing the Chi-square (CHI) feature selector as a filter step to discard less discriminative features. In the subsequent wrapper step, a novel algorithm that combines the fast global search capability of the GA with the positive feedback mechanism of ant colony optimization (ACO) is proposed. To validate the proposed approach, a series of experiments are conducted using the Reuters-21578 *corpus*, and the results are compared against other well-known techniques. The evaluation outcomes demonstrate that their method outperforms the alternative approaches in the majority of cases, highlighting its effectiveness.

In *Nafis & Awang (2020)*, an embedded feature selection technique that combines Term Frequency-Inverse Document Frequency (TF-IDF) and support vector machine-recursive feature elimination (SVM-RFE) for text classification in unstructured and high-dimensional data is proposed. The proposed technique aims to assess the importance of features in high-dimensional text documents and enhance the efficiency of feature selection, leading to improved text classification accuracy. Initially, TF-IDF is employed as

a filter approach to measure the importance of features in the text documents. Subsequently, SVM-RFE is utilized in a backward feature elimination scheme to recursively eliminate insignificant features from the filtered feature subsets. The study conducts experiments using a benchmark repository of Twitter posts, where relevant features are extracted through pre-processing. The pre-processed features are then divided into training and testing datasets. Feature selection is performed on the training dataset by calculating TF-IDF scores for each feature, followed by SVM-RFE for feature ranking. Only the top-ranked features are selected for text classification using the SVM classifier. The experimental results demonstrate that the proposed technique achieves a remarkable accuracy of 98%, surpassing other existing techniques. In conclusion, the proposed technique effectively selects significant features in unstructured and high-dimensional text documents.

In *Behera et al. (2021)*, a hybrid approach is proposed that combines two deep learning architectures, namely convolutional neural network (CNN) and long short term memory (LSTM) using word embedding, for the sentiment classification of reviews across various domains. The suggested Co-LSTM model aims to achieve two primary objectives in sentiment analysis. Firstly, it offers high adaptability for analyzing large-scale social data while considering scalability. Secondly, unlike conventional machine learning methods, it is not limited to a specific domain. The experimental evaluation involves training the model on four diverse review datasets to capture different types of dependencies commonly found in posts. The experimental results indicate that the proposed ensemble model surpasses other machine learning approaches in terms of accuracy and other performance metrics for sentiment analysis task.

An Ontology and Latent Dirichlet Allocation (OLDA)-based approach for sentiment classification is proposed by *Ali et al. (2019)*. The system, developed using Web Ontology Language and Java, retrieves transportation content from social networks and applies OLDA to extract meaningful information by generating topics and features. Additionally, word embedding techniques are employed to represent documents, and lexicon-based approaches are utilized to enhance the accuracy of the word embedding model. The proposed approach is evaluated using machine learning classifiers, demonstrating an accuracy of 93% and confirming its effectiveness in sentiment classification.

The effectiveness of numerical word representation models is investigated in *Stein, Jaques & Valiati (2019)* through experimentation and analysis. The application of various models and algorithms is examined to assess their performance. Classification models are trained using prominent machine learning algorithms such as fastText, XGBoost, SVM, and CNN. Noteworthy word embedding methods, including GloVe, word2vec, and fastText, are employed for generating word embeddings. Publicly available data is utilized for training and evaluation, employing measures specifically suitable for the hierarchical context. F1 score of 0.893 is achieved by fastText on a single-labeled version of the RCV1 dataset. The analysis indicates that employing word embeddings and their variations presents a highly promising approach for hierarchical text classification.

In the study (*Berahmand et al., 2024*), authors attempt to furnish such an exhaustive survey, commencing with an elucidation of the foundational principles underlying

conventional autoencoders and their developmental trajectory. Subsequently, a systematic classification of autoencoders based on their architectural configurations and operational principles is proffered, meticulously scrutinizing and deliberating upon associated models. Additionally, a comprehensive review of autoencoder applications across diverse domains such as machine vision, natural language processing, complex network analysis, recommender systems, speech processing, anomaly detection, among others, is undertaken. Finally, the current constraints inherent in autoencoder algorithms are synthesized and prospective avenues for future advancements within this realm are deliberated upon.

## Feature selection with heuristic optimization-based techniques

In *Purushothaman, Rajagopalan & Dhandapani (2020)*, a technique demonstrating a high level of maturity in convergence rate and requiring minimal computational time is proposed, thereby avoiding entrapment in local minima within a low-dimensional space. The input comprises text data, which undergoes initial preprocessing steps within the document. Subsequently, text feature selection is carried out by identifying local optima from the text document, and then the best global optima are selected from these local optima using a hybrid GWO-GOA approach. Additionally, the selected optima are subjected to clustering through the utilization of the Fuzzy c-means (FCM) clustering algorithm. This algorithm enhances reliability and minimizes computational time expenditure. The proposed algorithm is tested using eight datasets, exhibiting efficient performance. Evaluation metrics, including accuracy, precision, recall, F-measure, sensitivity, and specificity, are employed to evaluate text feature selection and text clustering. Comparisons with various other algorithms reveal that the proposed methodology exhibits superior quality, surpassing GWO, GOA, and the hybrid GWO-GOA algorithm, with an efficiency rate of 87.6%.

In *Moh'd Mesleh & Kanaan (2008)*, the implementation of a text classifier utilizing the support vector machine (SVM) model for Arabic articles has been undertaken. Furthermore, a novel feature subset selection (FSS) method based on Ant Colony Optimization (ACO) and Chi-square statistic has been developed and implemented. The proposed ACO-Based FSS method incorporates Chi-square statistic as heuristic information and leverages the efficacy of the SVM classifier as a guide to enhance feature selection for each category. In comparison to six prevailing FSS methods, the ACO-Based FSS algorithm demonstrates superior effectiveness in text classification. The evaluation utilizes an internally developed *corpus* for Arabic text classification, comprising 1,445 documents that have been independently categorized into nine distinct classes. The experimental results are presented in terms of macro-averaging precision, macro-averaging recall, and macro-averaging F1 measures.

In *Jain & Kashyap (2023)*, the sentiment categorization of text is presented, encompassing pre-processing, feature vector extraction, feature selection, and classification procedures. The utilization of the grey wolf technique is employed for feature selection, extracting salient text features from the feature vector. Additionally, a deep learning neural network is employed for classifying the text into positive and negative

sentiments. The proposed model is validated using publicly available datasets from Twitter (sentiment140) and movie reviews (IMDb). Remarkable achievements are observed, with precision, accuracy, recall, and F-measure reaching the highest levels of 94.16%, 95.81%, 92.87%, and 95.22%, respectively.

In *Hosseinalipour et al. (2021)*, the integration of the Black Widow Optimization (BWO) algorithm into a binary algorithm for addressing discrete problems is initially presented. The utilization of opposition-based inputs is employed to expedite the attainment of optimal solutions. Furthermore, to tackle the property selection problem, which is characterized by multiple objectives, the algorithm is transformed into a multi-objective algorithm. To evaluate the performance of the proposed method, 23 well-known functions are utilized. The results demonstrate favorable outcomes. Additionally, a practical example involving the application of the proposed method to various emotion datasets is considered. The findings indicate the remarkable efficacy of the proposed method in the domain of text psychology.

A new method called meta-heuristic feature optimization (MHFO) for data security in the campus workplace with robotic assistance is proposed in *Gong, Dinesh Jackson Samuel & Pandian (2021)*. Firstly, the terms of the space vector model are mapped to the concepts of data protection ontology, which enables the calculation of conceptual frequency weights by considering various weights assigned to each term. Moreover, based on the designs of the data protection ontology, the weight of theoretical identification is assigned. The combination of standard frequency weights and weights derived from the data protection ontology significantly reduces the dimensionality of functional areas. Additionally, this process integrates semantic knowledge. The results demonstrate that the advancement of the characteristics within this process leads to a notable improvement in secure text mining in the campus workplace. In conclusion, the experimental findings indicate that enhancing the features of the concept hierarchy structure process substantially enhances the data security of campus workplace text mining when robotic assistance is employed.

# PROPOSED FRAMEWORK

Portions of this text were previously published (https://arxiv.org/abs/2401.10270) as part of a preprint. MBO (*Duman, Uysal & Alkaya, 2012*) is a population-based metaheuristic algorithm that mimics the social interaction and movement patterns observed in bird populations. MBO technique recognized for its efficacy in solving complex combinatorial problems (*Hussain et al., 2019*; *Zhao, Wang & Zhang, 2019*; *Brezočnik, Fister & Podgorelec, 2018*). In this research, we harness the inherent strengths of MBO to introduce a novel and promising MBO as a feature selection algorithm. In other words, the proposed approach capitalizes on MBO's ability to iteratively refine solutions, optimizing feature subsets to maximize classification performance.

This section provides a comprehensive overview of our novel approach, detailing the incorporation of MBO into the feature selection.

## Migration birds optimization

The MBO model (*Duman, Uysal & Alkaya, 2012*) is inspired by the behavior of bird flocks during migration, specifically $N$ birds, with $N$ being an odd number, flying in a $V$-shaped

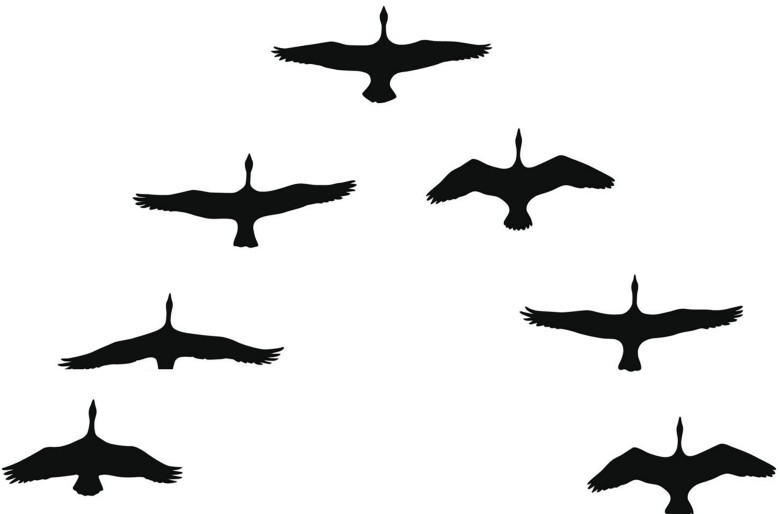

**Figure 1 V-shaped formation of seven birds (*Duman, Uysal & Alkaya, 2012*).**

formation (Fig. 1). Each bird represents a potential solution, undergoing refinement through the course of flights. A random bird is chosen as the leader, guiding the entire flock, while the remaining birds follow in two orderly lines, maintaining the *V*-shaped formation (Fig. 1). This innovative approach leverages the dynamics of a bird flock to enhance the exploration and exploitation capabilities of the optimization algorithm.

In Table 1, the detailed explanation of each parameter used in the study is given to provide a better understanding of the proposed framework. Algorithm 1 outlines the MBO, which receives an input of $M \times N$ data and the number of birds in the flock, and strives to enhance the classification outcome by eliminating less significant features.

In the initial step, it calculates the fitness of the input data, which is a way to see how well the data works within the context of different solutions (is discussed in "Fitness function" in detail), denoted as $F$ where $0 \leq F \leq 1$ (line 1 of Algorithm 1). In line 2, we assign the initial fitness $F$ value to 3 different variables in order to observe the improvement of the fitness value. More specifically, $F_i$ ($1 \leq F \leq 3$) retains the fitness value from the fitness values of the iteration $i$ steps earlier (line 20). Hence, when the difference between $F_1$ and $F_3$ reaches 0 (line 8), it indicates that the same result has been achieved in the last three steps. At this point, we conclude the algorithm, presuming that no additional improvement can be achieved.

At line 6, the flock is initialized with an odd number of birds. An illustration of a flock with seven birds is presented in Fig. 1. The bird positioned at the forefront, *i.e.*, at the corner of the *V*-formation, is designated as the leader of the flock. The flock iteratively engages in flight and landing cycles, wherein optimizations are applied to enhance the flock's performance. After completing a cycle, a new leader is chosen based on the bird with the highest fitness value, and the position of the current leader is replaced with the newly selected leader (line 19) while the positions of the remaining birds unchanged. Subsequently, the flock initiates another flight, following the same sequence of steps, and

**Table 1  The explanation of parameters used in the study.**

| Symbols | |
| --- | --- |
| M | # of feature extracted from data |
| N | # of instance |
| $M'$ | # of feature of subset |
| F | Fitness of input data |
| $F_i$ | Fitness value of iteration i |
| $F_{max}$ | Fitness value of $B_{max}$ |
| $B_{max}$ | The best bird ever found |
| $b_1$ | The leader of the flock |
| $b_i$ | Birds in flock |
| $b_i'$ | First best of solution |
| $b_i''$ | Second best of solution |
| $b_i'''$ | Third best of solution |
| $f_i$ | Selected feature |

this process continues iteratively until the flock reaches its desired destination. This iterative process contributes to the ongoing refinement of the solutions explored by the bird flock.

We depict the steps carried out between lines 8–22 in Fig. 2. Each step within the upper large box of Fig. 2 corresponds to lines 11 in Algorithm 1. In these steps, we make some changes in the solutions which will be explained in detail in the following paragraph, with the goal of discovering more optimized solutions, referred to as birds, throughout the flight. The number of changes is decided by the number of *change* variable as given in line 9. For instance, if the *change* is 3, we make three changes in the current solution. If an optimization could have been made, the global best fitness value (lines 14–17) is updated. After 10 steps, a number determined through previous experiments, we reorder the flock by replacing the bird with the highest fitness value with the leader (line 19). The algorithm terminates when either no optimization is possible, *i.e.*, the last three steps produces the same results, or it reaches max number of iterations, that is 100 (line 8). Then, the best solution found so far $F_{max}$ is returned as output.

Each flock fly operation (line 10), *i.e.*, shown as arrows in the Fig. 2, corresponds to the operations given in Fig. 3. We, for simplicity, elaborate a single iteration (line 10) with a hypothetical flock having five birds $\{b_1, b_2, b_3, b_4, b_5\}$ where $b_1$ is the leader, $b_2$ and $b_3$ follows the leader and $b_4$ and $b_5$ follows $b_2$ and $b_3$, respectively, in V-shaped formation as illustrated in Fig. 3A. In Fig. 3B, every bird $b_i$ generates a unique set of neighbors $B_i$ by creating similar solutions as itself through a small number of changes (neighbor generation). Then, in Fig. 3C, the leader $b_1$ replaces itself with the best solution selected from the set $\{b_1\} \cup B_1$ and selects two more best solutions, *i.e.*, the second $b_1''$ and the third best $b_1'''$ solutions, and $b_1''$ is added to its left set of birds and the $b_1'''$ is added to its right set of birds. Thus, the set of birds on the left side of the leader becomes $\{b_2\} \cup B_2 \cup \{b_1''\}$ and the set of birds on the right side of the leader becomes

**Algorithm 1** MBO algorithm.

**input**: $MxN$ data, flockSize,

**output**: $M'xN$ data where $0 < M' \leq M$

1: $F = computeFitness(data)$

2: $F_1,\ F_2,\ F_3 \leftarrow F,\ F,\ F$

3: $F_{max} \leftarrow F$

4: $B_{max} \leftarrow data$

5:

6: $flock \leftarrow initializeFlock(data, flockSize)$

7: $counter \leftarrow 0$

8: **while** $(counter < 3\ ||\ F1 - F3 == 0)\ \&\&\ (counter < 100)$ **do**

9:  $change \leftarrow calcChangeCount(count)$

10:  **for** $i$ in $1..10$ **do**

11:   $flock.fly(M, change)$

12:   $Bird_{best} \leftarrow flock.findBestBird()$

13:   $F_{best} \leftarrow Bird_{best}.getFitness()$

14:   **if** $F_{best} > F_{max}$ **then**

15:    $F_{max} \leftarrow F_{best}$

16:    $B_{best} \leftarrow Bird_{best}$

17:   **end if**

18:  **end for**

19:  $flock.reorder()$

20:  $F_1,\ F_2,\ F_3 \leftarrow F_{max},\ F_1,\ F_2$

21:  $counter \leftarrow counter + 1$

22: **end while**

23: **return** $B_{max}$

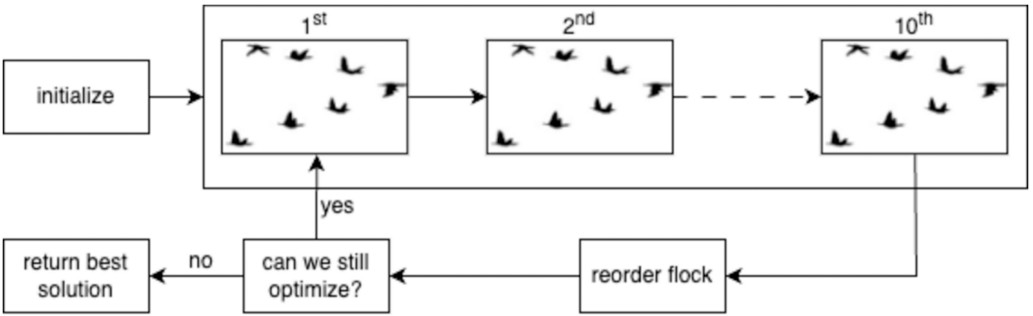

**Figure 2** Overview of MBO algorithm (*Duman, Uysal & Alkaya, 2012*).

| $b_1$ | |
|---|---|
| $b_2$ | $b_3$ |
| $b_4$ | $b_5$ |

**(a)** A flock with 5 birds in V-shaped formation as $b_1$ is being the leader

| $\{b_1\} \cup B_1$ | |
|---|---|
| $\{b_2\} \cup B_2$ | $\{b_3\} \cup B_3$ |
| $\{b_4\} \cup B_4$ | $\{b_5\} \cup B_5$ |

**(b)** Set of candidate birds as replacement for the $b_i$, created by generating neighbors $B_i$ of the bird $b_i$

| $b_1'$ | |
|---|---|
| $\{b_2\} \cup B_2 \cup \{b_1''\}$ | $\{b_3\} \cup B_3 \cup \{b_1'''\}$ |
| $\{b_4\} \cup B_4$ | $\{b_5\} \cup B_5$ |

**(c)** $b_1'$ is chosen as the new leader, the second $(b_1'')$ and third best $(b_1''')$ birds of $\{b_1\} \cup B_1$ set, is added to the set of left and right birds (to the followers), respectively.

| $b_1'$ | |
|---|---|
| $b_2'$ | $b_3'$ |
| $\{b_4\} \cup B_4 \cup \{b_2''\}$ | $\{b_5\} \cup B_5 \cup \{b_3''\}$ |

**(d)** $b_2'$ and $b_3'$ are selected as replacements for their positions, and their respective second-best birds, $b_2''$ and $b_3''$, are included in the set of followers birds.

| $b_1'$ | |
|---|---|
| $b_2'$ | $b_3'$ |
| $b_4'$ | $b_5'$ |

**(e)** $b_4'$ and $b_5'$ are selected as replacements for their positions out of sets given in Figure 3d.

**Figure 3 An example of birds replacements in the flock by using neighbor states.**

$\{b_3\} \cup B_3 \cup \{b_1'''\}$ (Fig. 3C). In the fourth step (Fig. 3D), the birds $b_2$ and $b_3$ replace themselves with the best solution selected from the sets $\{b_2\} \cup B_2 \cup \{b_1''\}$ and $\{b_3\} \cup B_3 \cup \{b_1'''\}$, respectively. Subsequently, both $b_2$ and $b_3$ selects their second best solutions, *e.g.*, $b_2''$ and $b_3''$, and they are added to the left and right set of birds, respectively. In the last step, the birds $b_4$ and $b_5$ replaces themselves with the best solutions selected from the sets $\{b_4\} \cup B_4 \cup \{b_2''\}$ and $\{b_5\} \cup B_5 \cup \{b_3''\}$, respectively. Finally, the new flock contains $\{b_1', b_2', b_3', b_4', b_5'\}$ birds.

## Modeling the feature selection as heuristic problem

In this section, we are not solely approaching the feature selection problem as an MBO problem. Instead, we present a generic modeling approach. This approach enables the utilization of various heuristic optimization techniques, provided they meet the specified requirements, for the feature selection.

Traditional heuristic methods typically require two key components: a *fitness function* and a *neighbor generation function*. The fitness function serves as a crucial evaluation metric, quantifying the quality or effectiveness of a potential solution within the problem space. It assigns a numerical value to each solution, indicating its performance with respect to the optimization goal. On the other hand, the neighbor generation function determines how neighboring solutions are generated. This function plays an important role in the exploration of the solution space, influencing the search strategy by defining the set of neighboring solutions considered for potential improvements. Together, the fitness function and the neighbor generation function form the core elements guiding the heuristic search process, allowing the algorithm to iteratively evaluate and refine solutions in the pursuit of optimal outcomes.

Therefore, we first introduce how we define a solution for feature selection problem and then present novel fitness and neighbor approaches for the problem in the following sections.

**Table 2 A solution for a hypothetical data having 10 features.**

| $f_0$ | $f_1$ | $f_2$ | $f_3$ | $f_4$ | $f_5$ | $f_6$ | $f_7$ | $f_8$ | $f_9$ |
|-----|-----|-----|-----|-----|-----|-----|-----|-----|-----|
| 1 | 1 | 0 | 0 | 1 | 0 | 0 | 1 | 1 | 0 |

### Solution

The data is given in $N \times M$ matrix format, where $N$ is the number of rows, *e.g.*, textual data; news or tweets in this study, and $M$ is the number of features extracted from those texts. Based on this format, a *solution* $N \times M'$ is represented as a subset of $M'$ features where $0 < M' < M$.

We represent all the solutions with an M-length vector with 0 and 1 s such that $i$-th position of the vector represents whether the respective feature is chosen or not. For example, consider the vector given in Table 2 for hypothetical data having 10 features. This vector represents the reduced data that is constructed by selecting the features $f_0, f_1, f_4, f_7,$ and $f_8$.

This reduced data, in fact, represents a $N \times M'$ tabular data as given in Table 3.

This vector representation is chosen for storage efficiency not to copy the whole data repeatedly considering that thousands of solutions will be generated during the process.

### Neighbor generation

A neighbor state is generated by utilizing a solution and modifying it through the addition or removal of a small number of features. This operation is done randomly selecting positions in the feature vector and then applying the function given in Eq. (1) on each position. That is, converting 1 s into 0 and 0 s into 1. Therefore, a neighbor state can be considered as similar version of the state which it originates. This dynamic approach allows for the exploration of nearby solutions within the problem space. For instance, the feature vector given in Table 4, can be generated from the solution given in Table 2 using the function (Eq. (1)). Assuming the randomly chosen features are $f_0$ and $f_9$, then, the neighbor state is generated by excluding the feature $f_0$ ($F(f_0) = F(1) = 0$) and including the feature $f_9$ ($F(f_1) = F(0) = 1$).

$$F(f_i) = 1 - f[i] \tag{1}$$

for all randomly chosen features $f_i$ where f is the feature vector and f[i] is the i-th element of the feature vector.

In the MBO algorithm, the initial flock comprises seven birds, with each bird having five neighbors and undergoing ten flaps per iteration, while the overlap factor is set to 1. The selection of these parameters is guided by the goal of optimizing the equilibrium between solution accuracy and computational efficiency during the optimization process. Augmenting the number of birds and neighborhoods typically enhances solution accuracy by fostering both exploration and exploitation; however, this augmentation is accompanied by increased computational time due to heightened computational demands. Similarly, increasing the iteration count can initially enhance outcomes, yet we constrain it to forestall diminishing returns and minimize unnecessary runtime. These strategic

**Table 3 The corresponding reduced data for the solution given in Table 2.**

|  | $f_0$ | $f_1$ | $f_4$ | $f_7$ | $f_8$ |
|---|---|---|---|---|---|
| $r_0$ | $v_{00}$ | $v_{01}$ | $v_{04}$ | $v_{07}$ | $v_{08}$ |
| $r_1$ | $v_{10}$ | $v_{11}$ | $v_{14}$ | $v_{17}$ | $v_{18}$ |
| $r_2$ | $v_{20}$ | $v_{21}$ | $v_{24}$ | $v_{27}$ | $v_{28}$ |
| .... |  |  |  |  |  |
| $r_N$ | $v_{N0}$ | $v_{N1}$ | $v_{N4}$ | $v_{N7}$ | $v_{N8}$ |

**Table 4 A neighbor state generated by excluding $f_0$ and including $f_9$ for the solution given in Table 2.**

| $f_0$ | $f_1$ | $f_2$ | $f_3$ | $f_4$ | $f_5$ | $f_6$ | $f_7$ | $f_8$ | $f_9$ |
|---|---|---|---|---|---|---|---|---|---|
| 0 | 1 | 0 | 0 | 1 | 0 | 0 | 1 | 1 | 1 |

choices are pivotal in maintaining a delicate balance between solution precision and computational constraints, thereby facilitating effective exploration of the solution space while mitigating excessive computational load.

### Fitness function

We define the fitness function to assess the efficacy of the reduced data, quantifying it as the ratio of correctly classified instances. To do so, we employ the cross-validation technique using well-known Python library scikit-learn (*Pedregosa et al., 2011*) to construct a classification model (Models) using the reduced data, *i.e.*, the solution.

More specifically, the fitness function $f$ given in Eq. (2) takes a classification algorithm $C$, a number $K$ and data $D$ as inputs, and computes $K$-fold cross-validation.

$$f(C, K, D) = cross\_validation(C, K, D) \qquad (2)$$

where C is the classification algorithm, K is a number for K-fold cross-validation and D is the data. The result of the *cross_validation* function is the average ratio of correctly classified instances over all folds, providing a measure of the classification algorithm performance. Then, a superior solution is indicated by a higher ratio of correctly classified instances.

### MBO as feature selection

MBO can be regarded as a conventional heuristic optimization technique, incorporating standard practices for fitness functions and neighbor utilization. Therefore, MBO is well-suited to serve as an effective approach for addressing the feature selection problem. This section outlines our utilization of MBO to tackle the feature selection problem.

We start with employing Info Gain (*Quinlan, 1986*; *Karegowda, Manjunath & Jayaram, 2010*) feature selection algorithm to reduce the feature set to a manageable size. This initial reduction is essential as MBO's efficiency is exponentially influenced by the number of features. The reduced data resulting from Info Gain serves as an input for MBO. Subsequently, we feed this reduced data into MBO, allowing heuristic operations to

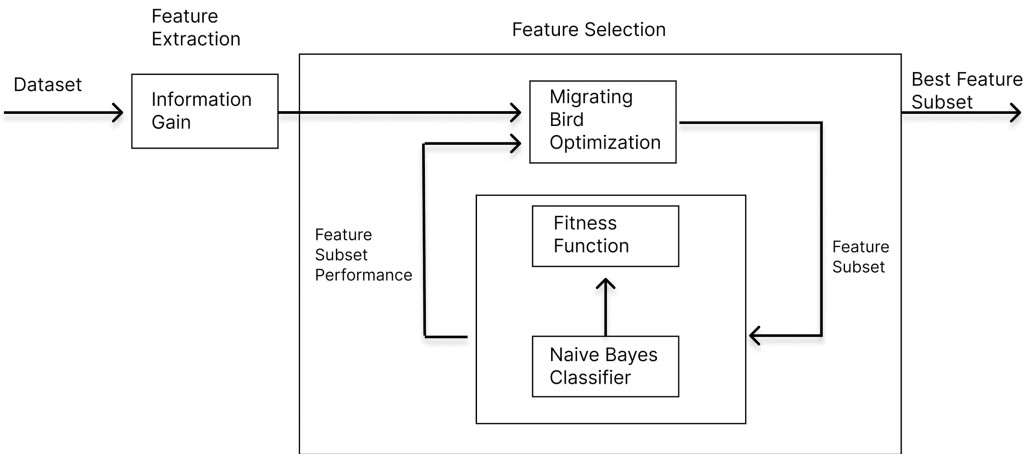

**Figure 4  Block diagram of the MBO algorithm.**     

identify and select the most beneficial features among the remaining set. This approach ensures that, in the worst-case scenario, our selection aligns with the features identified by Info Gain. The Info Gain algorithm selects all features contributing to classification, unless the total selected features surpasses 2,500 which is a limitation imposed for performance reasons of MBO. We, then, use the reduced data as input for MBO.

We already explained the details of Algorithm 1 in this section. It takes this reduced $M \times N$ data as input and produces a $M' \times N$ solution by selecting $M'$ best features out of initial $M$ features. In Fig. 4, the block diagram of MBO algorithm is given for the purpose of demonstrating the feature selection process.

## EXPERIMENTS

In this section, we evaluate the effectiveness of MBO in addressing the feature selection problem. The experiments are designed to compare its performance against well-known feature selection algorithms. For this purpose, in the first phase of experimentation, we attempt to select the best classifier to complement the MBO framework. This classifier selection is important because the fitness function (fitness function) determines the value of the solution. The computed fitness value, derived from this function, is subsequently employed to compare solutions and decide the best solution among others. In the second phase of experiments, we compare the performance of MBO with other well-known feature selection algorithms. The goal of these experiments is to figure out how well MBO can be applied in real-world problems by understanding its strengths and where it fits in the world of feature selection.

### Data

In the experiments, we utilized a diverse set of data, each characterized by specific features, instances, and classes. The details of the data are outlined in Table 5. The table presents key information regarding five different data: 20News-18828 (*Mitchell, 1997*), aahaber (*Tantuğ, 2010*), hurriyet (*Kilimci & Akyokus, 2018*), mininews (*Kilimci & Akyokus, 2018*), and webkb4 (*Craven et al., 1998*).

In Table 5, the "no of feature" column indicates the number of features or attributes associated with each data. The "no of instances" column denotes the total number of instances or data points in each data. The "no of classes" column represents the number of classes or categories present in each data. The "avg. word count per instance" column provides the average word count per data instance whereas the "avg. word length" column indicates the average length of words in each data. And lastly, the language column gives the language of the text used in the study. For instance, the data "hurriyet" contains 71,071 features, 6,006 instances, and six classes (or categories). The average word count per instance and average word length in data are 158.3 and 7.1, respectively. The text language is in Turkish.

We initially acquired the data in its raw textual format. During the data preprocessing phase, we employ tokenization to break down the text into individual units, followed by the removal of stop words to enhance the efficiency of subsequent analyses. Then, we utilized Term Frequency-Inverse Document Frequency (TF-IDF) (*Joachims, 1997*) to evaluate the importance of terms in a document relative to a collection of documents. Finally, we construct feature vectors, representing the processed data in the feature vector format to further computational operations.

These data with varying characteristics provide a rich set of examples for our experiments, allowing us to explore and analyze the performance of different models across diverse data scenarios.

## Models

In this section, the models employed in this research are briefly introduced. In the scope of this work, Information Gain (*Quinlan, 1986*), PSO (*Abualigah, Khader & Hanandeh, 2018*) and MBO algorithms are assessed as feature selection techniques while naïve Bayes, Decision Tree, multilayer perceptron, and support vector machine, are evaluated as machine learning techniques. In each model creation, use used five-fold cross-validation technique.

The naïve Bayes algorithm, widely employed for classification tasks, operates based on the principles of Bayes' theorem (*Dale, 2005*; *Xu, 2018*), assuming conditional independence of features given the class label. The algorithm's functioning entails the preparation of labeled data for training, estimation of probabilities using Bayes' theorem, and prediction of class labels based on the highest probability.

Decision Tree algorithm (*Quinlan, 1986*; *Charbuty & Abdulazeez, 2021*) involves recursively partitioning the input data based on feature values to create a hierarchical structure resembling a tree. At each internal node of the tree, a decision rule is applied to determine the optimal feature to split the data, maximizing the homogeneity within each resulting subset. The process continues until a stopping criterion is met, typically when the data subsets reach a predetermined purity threshold or a specified depth is reached. The resulting tree can then be used to make predictions for new, unseen instances by traversing the tree from the root node to a leaf node, following the decision rules at each internal node.

**Table 5 Characteristics of the data used in the experiments.**

| Data | No. of features | No. of instances | No. of classes | Avg. word count per instance | Avg. word length | Language |
|------|------|------|------|------|------|------|
| 20News-18828 | 113,241 | 18,828 | 20 | 272.5 | 5.3 | English |
| aahaber | 48,983 | 20,000 | 8 | 33.6 | 6.5 | Turkish |
| hurriyet | 71,071 | 6,006 | 6 | 158.3 | 7.1 | Turkish |
| mininews | 35,991 | 2,000 | 20 | 311.0 | 6.3 | English |
| webkb4 | 41,818 | 4,199 | 4 | 281.0 | 6.3 | English |

In multilayer perceptron (MLP), information moves forward from one layer to the next, starting with the input nodes, then progressing through hidden layers (which can be one or more layers), and finally reaching the nodes in the output layer (*Tuan Hoang et al., 2021*). These networks are made up of a mix of simpler models known as sigmoid neurons. MLPs consist of many nodes, and sigmoidal functions are commonly used as activation functions. MLPs have the ability to grasp the complex and nonlinear decision boundaries from data. They typically have one input layer, one output layer, and multiple hidden layers positioned between the input and output layers. The inclusion of more hidden layers enables MLPs to understand more intricate nonlinear relationships between the input and output.

Particle Swarm Optimization (PSO), introduced by *Kennedy & Eberhart (1995)*, represents an emergent population-based meta-heuristic. This method emulates social behaviors, specifically the phenomenon of birds flocking towards a promising position to achieve precise objectives within a multidimensional space. Similar to evolutionary algorithms, PSO conducts searches utilizing a population, referred to as a swarm, composed of individuals termed particles. These particles undergo iterative updates to enhance their positions. In the pursuit of discovering the optimal solution, each particle adjusts its search direction based on two influential factors: its individual best previous experience and the collective best, incorporating both cognitive and social aspects. It is characterized by its straightforward implementation requiring few parameters, and it has found extensive application in addressing optimization problems, including those related to feature selection (*Lin et al., 2008*).

Support vector machines algorithm (*Vapnik, 2013*; *Pisner & Schnyer, 2020*) involves identifying an optimal hyperplane in a high-dimensional feature space to separate different classes of data points. SVM achieves this by maximizing the margin, which is the distance between the hyperplane and the nearest data points from each class, known as support vectors. This optimal hyperplane is determined by solving a convex optimization problem that aims to minimize classification errors while maximizing the margin.

Information Gain revolves around quantifying the relevance of features in a data by measuring their contribution to the reduction of uncertainty during the classification process (*Quinlan, 1986*; *Azhagusundari & Thanamani, 2013*). It calculates the information gain by evaluating the difference in entropy (or impurity) before and after the feature is

considered. Features with higher information gain are deemed more informative and are thus selected for further analysis.

## RESULTS

In this section, we share the experimental results. We also present a statistical significance test thereby utilizing two tailed Student's t-Test particularly when results of naïve Bayes-based approaches (namely, raw data, Information Gain, MBO, and PSO) are close to each other (*e.g.*, about 2–3% difference). Whether the probability associated with Student's t-Test is lower, we consider the difference is statistically significant if it is arranged as $\alpha = 0.05$ significance level. In the first phase, we aimed to identify the most suitable classification algorithm for integration into our proposed approach. For this purpose, we employed a set of diverse classifiers, namely Decision Tree, multilayer perceptron, naïve Bayes, and support vector machine. The objective of this phase was to identify the optimal classifier that aligns most effectively with the requirements of our algorithm. However, we did not choose any complex classification algorithms that require long running time such as deep learning techniques (*Liang et al., 2017*; *Kilimci & Akyokus, 2018*) since our proposed approach needs to create classification models repeatedly.

Note that the average feature count across the data (Table 5) is approximately 62,221. Given the significant impact of feature count on the performance of our fitness function, we took measures to enhance efficiency. To do so, we applied the Information Gain algorithm to the raw data before starting the MBO algorithm. That is, the input data for the MBO algorithm is derived from the output of the Information Gain process. To mitigate computational demands, we imposed a cap on the maximum number of features, limiting it to 2,500. It is important to emphasize that, for efficiency considerations, we utilized Information Gain (models) as a representative of our raw data, rather than processing the entire data directly. This strategic approach aims to balance computational efficiency with the essential need to optimize the performance of our fitness function.

We share the first set of results in Table 6. For each data, we applied MBO with different classifiers, reporting the correctly classified ratio of final reduced data. We applied a 10-h threshold to ensure timely completion of the experiments, while also accounting for practical considerations and constraints. Therefore, if an experiment exceeded this threshold, the best result computed within the timeframe is reported.

In Table 6, results from 20 different configurations, involving four classifiers and five data, are presented. The overall average correctly classified percentages are 37.3, 66.9, 83.1, and 35.9 for Decision Tree, multilayer perceptron, naïve Bayes, and support vector machines, respectively. Notably, the highest percentages are observed for multilayer perceptron and naïve Bayes. To further investigate, we compared their results individually. Specifically, the minimum and the maximum correctly classified percentages is 47.4 and 80.1 for multilayer perceptron, respectively and 76.4 and 91.1 for naïve Bayes, respectively.

Based on the result of the initial set of experiments, we have chosen to incorporate naïve Bayes as the classifier for the fitness function within the MBO algorithm. Henceforth, we refer to this integrated approach as MBO-NB.

**Table 6 Comparing classifier integrations of MBO in terms of correctly classified percentage of the reduced data.**

| Data | Decision tree | Multilayer perceptron | Naive bayes | Support vector machines |
|---|---|---|---|---|
| 20News-18828 | 4.4 | 47.4 | 83.0 | 5.3 |
| aahaber | 32.2 | 76.1 | 79.7 | 44.0 |
| hurriyet | 43.0 | 62.0 | 76.4 | 42.7 |
| mininews | 30.1 | 69.1 | 85.5 | 48.2 |
| webkb4 | 76.9 | 80.1 | 91.1 | 39.1 |

In the second set of experiments, we attempted to compare the effectiveness of MBO-NB's output data with its raw data, and reduced data by Information Gain and PSO. For a fair comparison, our envisioned approach is as follows: users initially experiment with several classifiers using a five-fold cross-validation. Subsequently, only the best-performing classifier is selected and applied to the test data. Thus, our reporting focuses solely on the results obtained using the best classifier out of naïve Bayes, multilayer perceptron and Decision Tree. However note that this is for the evaluation of the reduced data, MBO-NB still uses naïve Bayes as internal classifier.

We present these results in Table 7.

Note that PSO could not succeed to reduce the data within the time limitation. Therefore, to make a fair comparison, we exclude the result for aahaber and present the statistics for the remaining data. That is, for instance, taking average of 20News-18828, hurriyet, mininews and webkb4 results. With this comparison, the average percentage for the comparable results are 84.8, 83.7, 87.6, and 80.4, for the raw data, Information Gain, MBO-NB, and PSO, respectively. In order to use all data results in the comparison, we exclude the PSO results and take the averages of all data results. The averages become 84.1, 82.9, and 86.7, for the raw data, Information Gain, and MBO-NB respectively. In both comparisons, except 20News-18828 data, MBO-NB outperforms better than other techniques, that is 80% of the setup. More precisely, MBO-NB improved the percentages by 3.3%, 4.6%, and 9.2%, compared to the raw data, Information Gain and PSO, respectively.

Additionally, we share the feature counts for both the raw data and the reduced data resulting from the feature selection algorithms in Table 8. On average, Information Gain reduced the number of features in the raw data from 62,221 to 2,089, accompanying a decrease of 1.3 percentages in the correctly classified percentages. This trade-off is made to address performance concerns, meaning that we begin with a less favorable initial data, that is reduced data by Information Gain, but achieve a better correctly classified percentage compared to raw data, *i.e.*, 3.3%.

The experiment was implemented using the Python programming language with the scikit-learn library due to its versatility, extensive libraries for machine learning and optimization, and widespread adoption in the research community. Python offers a robust platform for developing and executing complex algorithms, making it well-suited for our

**Table 7 Comparing correctly classified percentages of the raw data and reduced data by Information Gain, MBO-NB and PSO.**

| Data | Raw data | Information gain | MBO-NB | PSO |
|---|---|---|---|---|
| 20News-18828 | 86.3 | 81.5 | 83.0 | 75.6 |
| aahaber | 81.0 | 79.7 | 82.8 | – |
| hurriyet | 74.5 | 76.4 | 82.0 | 72.5 |
| mininews | 87.8 | 85.5 | 94.2 | 86.0 |
| webkb4 | 90.7 | 91.3 | 91.3 | 87.5 |

**Table 8 Comparing the feature counts of the raw data and reduced data by Information Gain, MBO-NB and PSO.**

| Data | Raw data | Information gain | MBO-NB | PSO |
|---|---|---|---|---|
| 20News-18828 | 113,241 | 2,500 | 1,847 | 1,268 |
| aahaber | 48,983 | 2,500 | 1,696 | – |
| hurriyet | 71,071 | 2,500 | 1,352 | 967 |
| mininews | 35,991 | 467 | 193 | 224 |
| webkb4 | 41,818 | 2,422 | 1,484 | 1,153 |

study on feature selection in text classification tasks. The experiments were conducted on a machine equipped with an Intel Xeon CPU running at 2.30 GHz, 16 GB of RAM, and operating on 64-bit Ubuntu 18.04.

## DISCUSSIONS

Given the substantial feature count, on average 62,221, and its impact on our fitness function performance, we preprocessed the raw data using the Information Gain algorithm. This reduced the number of features from 62,221 to 2,089 on average, with a cap set at 2,500 for efficiency. Despite using Information Gain as a representative of raw data for efficiency considerations, our strategic approach balances computational efficiency with the essential need to optimize our fitness function's performance.

Our experiments show the effectiveness of MBO-NB, emphasizing its superiority in feature reduction over other techniques. The strategic integration of naïve Bayes as the internal classifier within MBO proves successful, offering a balanced solution for enhancing computational efficiency while maintaining optimal classification accuracy.

We also conduct an individual comparison of MBO and PSO, both utilizing heuristic approaches. Across all four setups, MBO-NB consistently outperforms PSO by an average of 6.9%.

## CONCLUSIONS

In this study, we evaluated the effectiveness of the Migration Birds Optimization (MBO) algorithm for the feature selection problem in text classification tasks. Through a designed series of experiments, we systematically investigated the performance of MBO in comparison to established feature selection algorithms, with a particular emphasis on

classifier selection and subsequent assessments. The primary objective of our research was to address the inherent challenges associated with feature selection in text classification by proposing an innovative methodology. Motivated by the shortcomings of existing techniques in effectively handling extensive feature sets, our study aimed to develop a robust and efficient solution that enhances both computational efficiency and classification accuracy.

We evaluated the effectiveness of the Migration Birds Optimization algorithm for the feature selection problem. Through a series of carefully designed experiments, we explored the performance of MBO in comparison to established feature selection algorithms, with a particular focus on classifier selection and subsequent assessments. The initial phase of experimentation led us to the integration of naïve Bayes as the fitness function classifier within the MBO algorithm. This decision was informed by an analysis of the results obtained from different classifiers in terms of classification accuracy. The selection of naïve Bayes aligns with the iterative nature of our proposed approach, emphasizing the need for repeated creation of classification models. Subsequent experiments, comparing MBO-NB with raw data, Information Gain, and Particle Swarm Optimization (PSO), revealed promising outcomes. MBO-NB consistently outperformed raw data and Information Gain, demonstrating its robustness in feature selection tasks. The comparative analyses showcased its superiority over PSO which is also a heuristic approach, reinforcing the efficacy of the proposed approach. Furthermore, the analysis of the relationship between raw data and Information Gain-reduced data highlighted the potential of MBO-NB to enhance accuracy, presenting a viable alternative for real-world applications.

MBO-NB offers several distinct advantages compared to traditional feature selection methods:

- Enhanced computational efficiency: By integrating Migrating Birds Optimization (MBO) with naïve Bayes as an internal classifier, MBO-NB efficiently handles extensive feature sets, significantly reducing computational overhead. This streamlined approach allows for faster processing times, making it particularly suitable for large-scale text classification tasks.

- Improved classification accuracy: Through extensive experiments, we have demonstrated the superior effectiveness of MBO-NB in feature reduction compared to other existing techniques. By strategically reducing the feature count while maintaining or even improving classification accuracy, MBO-NB outperforms traditional approaches, resulting in more reliable and precise classification outcomes.

- Comprehensive solution: The successful integration of naïve Bayes within the MBO framework offers a comprehensive and well-rounded solution to the feature selection problem in text classification. MBO-NB leverages the strengths of both MBO and naïve Bayes, combining optimization capabilities with probabilistic modeling to achieve robust and effective feature selection.

- Consistent performance: In individual comparisons with Particle Swarm Optimization (PSO), MBO-NB consistently outperforms by an average of 6.9% across

four setups. This consistent performance highlights the reliability and efficacy of MBO-NB in improving classification accuracy compared to alternative methods.

By emphasizing these advantages, we aim to underscore the significance and novelty of our proposed methodology, MBO-NB, in addressing the challenges of feature selection in text classification tasks. We believe that these distinct benefits contribute to the advancement of text classification techniques and facilitate the development of more robust and efficient classification systems.

## Research limitations

Despite aforementioned significant contributions, this proposed framework has some limitations.

- Limited adaptability to diverse data and problem contexts: While MBO-NB demonstrates effectiveness in feature reduction and classification accuracy improvement, its adaptability to diverse datasets and problem contexts may be limited. The current approach may not be optimized for specific domains or may exhibit varying performance across different types of data.
- Dependency on preprocessing techniques: The success of MBO-NB relies on preprocessing techniques, such as the Information Gain algorithm, to strategically reduce the feature count. However, the efficacy of these preprocessing steps may vary depending on the nature and characteristics of the dataset. There may be scenarios where the chosen preprocessing techniques are not optimal, leading to suboptimal feature selection outcomes.
- Lack of robustness to noisy or imbalanced data: MBO-NB may face challenges in handling noisy or imbalanced datasets, which are common in real-world applications. The method may not adequately account for the presence of noise or imbalance in the data, potentially leading to biased or unreliable classification results.
- Limited exploration of advanced techniques: While MBO-NB integrates Migrating Birds Optimization with naïve Bayes, it may not leverage more advanced techniques, such as deep learning algorithms and text-based transformer approaches. These advanced methods have shown significant promise in text classification tasks and may offer opportunities for further improving the performance and robustness of MBO-NB.

By addressing these weaknesses and exploring potential areas for improvement, future research can enhance the robustness, versatility, and applicability of MBO-NB in text classification tasks. Additionally, further investigation into the integration of advanced techniques and the development of more adaptive and scalable methodologies can contribute to the advancement of feature selection methods in text classification.

## Potential future research

The results of this research emphasize the effectiveness of MBO-NB as a feature selection algorithm. The careful selection of classifiers, coupled with strategic preprocessing steps,

contributes to the algorithm's efficiency and performance. The findings presented in this study provide valuable insights into the application of MBO-NB in diverse scenarios, with implications for data-driven decision-making processes.

As we move forward, future research endeavors should thoroughly explore specific domains, examining the adaptability of MBO-NB to varying data and problem contexts. The insights gained from this research contribute to the changing nature of feature selection methodologies, opening avenues for further exploration and refinement. Furthermore, another potential area for future investigation could be the integration of deep learning algorithms and text-based transformer approaches within the MBO framework, alongside the incorporation of word embedding techniques, presents intriguing avenues for future investigation, promising advancements in the field of text classification.

Based on the limitation of this article and the computational results, we propose some potential directions for future research.

- Hybridization with other optimization algorithms: Investigating the potential benefits of hybridizing MBO with other optimization algorithms, such as genetic algorithms or simulated annealing, can be a promising direction. Hybrid approaches may leverage the strengths of multiple algorithms to overcome individual weaknesses and achieve superior feature selection outcomes in text classification.

- Integration with deep learning models: Considering the recent advancements in deep learning models for natural language processing, future research can explore the integration of MBO-based feature selection methods with deep learning architectures, such as convolutional neural networks (CNNs) or recurrent neural networks (RNNs). This integration can leverage the expressive power of deep learning models while harnessing the efficiency of MBO for feature selection.

- Adaptation to multimodal data: With the increasing availability of multimodal data, including text, images, and audio, future research can investigate the adaptation of MBO-based feature selection methods to handle multimodal classification tasks. This involves extending the methodology to select relevant features from diverse modalities and integrating them into a unified classification framework.

- Scalability and efficiency: Addressing the scalability and computational efficiency of MBO-based feature selection methods is crucial for handling large-scale text classification tasks. Future research can investigate strategies to parallelize and distribute the computation, optimize memory usage, and develop scalable implementations to accommodate massive datasets and high-dimensional feature spaces.

By exploring these potential research directions, future studies can advance the state-of-the-art in MBO-based feature selection methods for text classification, leading to more effective, efficient, and interpretable solutions for real-world applications.

### Funding
The authors received no funding for this work.

### Competing Interests
The authors declare that they have no competing interests.

### Author Contributions
- Cem Kaya conceived and designed the experiments, performed the experiments, analyzed the data, performed the computation work, prepared figures and/or tables, authored or reviewed drafts of the article, and approved the final draft.
- Zeynep Hilal Kilimci conceived and designed the experiments, analyzed the data, performed the computation work, authored or reviewed drafts of the article, and approved the final draft.
- Mitat Uysal analyzed the data, authored or reviewed drafts of the article, and approved the final draft.
- Murat Kaya performed the experiments, analyzed the data, prepared figures and/or tables, authored or reviewed drafts of the article, and approved the final draft.

### Data Availability
    The code and raw data are available in the Supplemental Files.

### Supplemental Information
Supplemental information for this article can be found online at http://dx.doi.org/10.7717/peerj-cs.2263#supplemental-information.

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
