# Peer review of "Migrating birds optimization-based feature selection for text classification"

_PeerJ Computer Science, doi:10.7717/peerj-cs.2263_

## Round 0.1 · original submission · Major Revisions

The reviewers have recognized the value of the submission but have identified several areas that require attention prior to publication.

Specifically, but not only:
- improvements are needed in the presentation to clearly emphasize the contribution's key aspects and its advantages over the current state of the art.
- the related work section requires expansion
- the description of the experimental workflow should improved; here, the authors are encouraged to consider making the source code of their experiments available

**Language Note:** PeerJ staff have identified that the English language needs to be improved. When you prepare your next revision, please either (i) have a colleague who is proficient in English and familiar with the subject matter review your manuscript, or (ii) contact a professional editing service to review your manuscript. PeerJ can provide language editing services - you can contact us at [email protected] for pricing (be sure to provide your manuscript number and title). – PeerJ Staff

·

Basic reporting

This paper introduces a compelling concept, a novel approach leverages Migrating Birds Optimization (coupled with Naive Bayes as an internal classifier to address feature selection challenges in text classification having large number of features.. However, there are areas that could benefit from refinement before the manuscript attains publication readiness. I recommend that the authors meticulously address the following points in their revision:


1- The author should rewrite the abstract because it does not adequately describe your work.
2.The introduction should be revised to clearly present the main ideas and motivations behind the proposed research. Please ensure that the research question and motivation of the proposed study are clearly stated. It is important to cover the research gap adequately.
3. How was the experiment implemented, with MATLAB or Python? Will the code be publicly available?
4. Add a table of used symbols in the paper to improve readability.
5.It will be worth mention if the author can state the advantages of the chosen methods against others.
6. The equation variables must be described in all equations. Also, describe the presence of the equation and its action based on processing the data. Avoid undefined variables in the equation.
7- the authors should analyse how to set the parameters of the proposed methods in the framework. Do they have the “optimal” choice?
8. Section experiment, it would be good to have more information about how experiments have been conducted. What tools/software has been used
9.Please provide a more detailed explanation of future work and address the weaknesses of the proposed method. Expanding on these aspects will provide a better understanding of the limitations of the current approach and shed light on potential areas for improvement.
10.The Literature citation is not adequate, and the related work should be discussed:
1 Autoencoders and their applications in machine learning: a survey
2. A Deep Semi-Supervised Community Detection Based on Point-Wise Mutual Information

Experimental design

'no comment

Validity of the findings

'no comment

Additional comments

'no comment

Reviewer 2 ·

Basic reporting

1. Clear and unambiguous, professional English used throughout. - Good
2. Literature references, and sufficient field background/context provided. - Some of the references are old try to use new references.
3. Professional article structure, figures, tables. Raw data shared. - Good
4. Self-contained with relevant results to hypotheses. - Add some tests to justify your result like the t-test.
5. Formal results should include clear definitions of all terms and theorems and detailed proofs. - Good.

Experimental design

1. Original primary research within Aims and Scope of the journal. - Yes
2. Research question well defined, relevant & meaningful. It is stated how research fills an identified knowledge gap. - yes
3. Rigorous investigation performed to a high technical & ethical standard. - Add a model architecture using a Picture. so that the reader can easily understand the process.
4. Methods described with sufficient detail & information to replicate.- Good

Validity of the findings

1. Impact and novelty not assessed. Meaningful replication encouraged where rationale & benefit to literature is clearly stated. - Good
2. All underlying data have been provided; they are robust, statistically sound, & controlled. - Good
3. Conclusions are well stated, linked to original research question & limited to supporting results. - Good

Additional comments

The paper is overall good, make the above changes.

Cite this review as

---

## Round 0.2 · accepted · Accept

With the new version of their article, the authors have answered all reviewers' comments. The work can now be accepted.

·

Basic reporting

The authors have considered my previous comments and revised the paper. I have no more suggestions.

Experimental design

The authors have considered my previous comments and revised the paper. I have no more suggestions.

Validity of the findings

The authors have considered my previous comments and revised the paper. I have no more suggestions.